

# Climatic Controls on Metabolic Constraints in the Ocean

Precious Mongwe[1], Matthew Long[2], Takamitsu Ito[3,] Curtis Deutsch[4], and Yeray Santana-Falcón[5]

[1]Southern Ocean Carbon Climate Observatory (SOCCO), CSIR, Cape Town, South Africa

[2]Oceanography Section, Climate and Global Dynamics Laboratory, National Center for Atmospheric Research, Boulder, CO, United States of America

[3]School of Earth and Atmospheric Sciences, Georgia Institute of Technology, Atlanta, Georgia United States of America

[4]Department of Geosciences, Princeton University, Princeton, NJ, United States of America

[5]CNRM, Université de Toulouse, Météo-France, CNRS, Toulouse, 31057, France

**Corresponding Author**: Precious Mongwe (pmongwe@csir.co.za)

## Abstract

Observations and models indicate that climate warming is associated with the loss of dissolved oxygen from the ocean. Dissolved oxygen is a fundamental requirement for heterotrophic marine organisms (except marine mammals) and, since the basal metabolism of ectotherms increases with temperature, warming increases organisms' oxygen demand. Therefore, warming and deoxygenation pose a compound threat to marine ecosystems. In this study, we leverage an ecophysiological framework and compilation of empirical trait data quantifying the temperature sensitivity and oxygen requirements of metabolic rates for a range of marine species ("ecotypes"). Using the Community Earth System Model Large Ensemble, we investigate how natural climate variability and anthropogenic forcing impact the ability of marine environments to support aerobic metabolisms on interannual to multi-decadal timescales. Warming and deoxygenation projected over the next several decades will yield a reduction in the volume of viable ocean habitat. We find that fluctuations in temperature and oxygen associated with natural variability are distinct from those associated with anthropogenic forcing in the upper ocean. Further, the joint temperature-oxygen anthropogenic signals emerges sooner than independently from natural variability. Our results demonstrate that anthropogenic perturbations underway in the ocean will strongly exceed those associated with the natural system; in many regions, organisms will be pushed closer to or beyond their physiological limits, leaving the ecosystem more vulnerable to extreme temperature-oxygen events.





## 1. Introduction

Dissolved oxygen ($O_2$) is a fundamental metabolic requirement for heterotrophic marine organisms, excluding marine mammals (Portner, 2002; Keeling et al., 2010; Tiano et al., 2014). $O_2$ is declining due to warming, a tendency long predicted by models (Keeling et al., 2010; Long et al., 2016; Oschlies et al., 2018) and recently found evident at the global scale in compilations of in situ observations (Schmidtko et al., 2017; Ito et al., 2017). Deoxygenation is driven by the direct effect of reduced oxygen solubility with warming compounded by buoyancy-induced stratification in the upper ocean, which weakens the ventilation-mediated supply of fresh oxygen to the ocean interior. While the full ecological impacts of ocean deoxygenation remain uncertain, it is clear that the physiological impacts of oxygen loss on marine organisms can be considered explicitly in the context of warming: basal metabolic rates for ectothermic organisms depend on ambient temperature and increase with warming (Gillooly et al., 2001); thus, higher temperatures impose additional demand for oxygen to sustain aerobic respiration (Deutsch et al., 2015). Consequently, as the ocean warms, even present-day oxygen distributions may be insufficient to meet the oxygen demands of organisms living near key physiological thresholds (Deutsch et al., 2022).

While model projections clearly demonstrate that warming and deoxygenation are consequences of human-driven climate change, it is important to recognize that natural climate variability also produces important fluctuations in these quantities. Indeed, evidence suggests that natural variability contributes to hypoxic events, such as those observed in the California Current, where fish and benthic-organism mortality has been associated with low-$O_2$ waters impinging on the continental shelf (Pozo Buil and Di Lorenzo, 2017; Howard et al., 2020). A clear understanding of how natural climate variability drives fluctuations in metabolic state and the associated implications for organisms is a critical context in which to view long-term climate warming. Given that the natural system is highly dynamic, climate change signals are often masked by decadal-scale variability (Ito and Deutsch, 2010). While numerous authors have considered detection and attribution of climate change for physical and biogeochemical variables (Rodgers et al., 2015; Long et al., 2016; Schlunegger et al., 2019), comparatively little attention has been devoted to explicitly characterizing the relative influence of natural and anthropogenic drivers of changes in the ocean's capacity to support aerobic life. In this study, we approach this challenge



by leveraging the concept of the Metabolic Index ($\Phi$) introduced by Deutsch et al. (2015). $\Phi$ is based on the notion that aerobic organisms can persist only where the ambient oxygen partial pressure ($p$O$_2$) is sufficient to meet the requirements of sustaining respiration. $\Phi$ incorporates an explicit representation of the dependence of metabolic oxygen demand on temperature, thus providing a framework to consider how joint oxygen and temperature variability constrain viable habitat in the ocean.

Many ocean organisms may already be under threat from deoxygenation (Hoegh-Guldberg and Bruno, 2010; Breitburg et al., 2018); however, ongoing climate-driven loss of oxygen raises important questions about the future of marine ecosystems: How will anthropogenic changes in dissolved oxygen and temperature impact the capacity of ocean habitats to support aerobic metabolism? What is the spatial and temporal distribution of changes in the ocean's metabolic state associated with climate variability? At what point can anthropogenic change in the ocean's metabolic state be distinguished from natural variability? This study addresses these questions using a combination of metabolic theory, a dataset set quantifying key physiological parameters for a collection of marine species adapted to specific environments ("ecotypes"), and the oxygen and temperature distributions simulated in the Community Earth System Model, version 1 Large Ensemble (CESM1-LE), which includes 34 members simulating ocean biogeochemistry under climate variability and change from 1920–2100 forced using historical data and the Representative Concentration Pathway Scenario 8.5 (RCP85) (Kay et al., 2015; Long et al., 2016).

This paper is organized as follows. Section 2 presents a brief overview of the relevant metabolic theory, the associated empirical datasets, and describes our approach to analysis. In Section 3 we present results quantifying the joint temperature-oxygen variability simulated in the CESM1-LE, evaluating the spatiotemporal structure of variability in marine ecotype habitat, including long-term trends based on the RCP8.5 scenario and time of emergence (ToE). The main outcomes of the results are synthesized in Section 4 and summarized in Section 5.

## 2. Datasets and methods





**2.1 Metabolic index**

Empirical studies measuring thermal tolerance and oxygen requirements in the laboratory on an
array of marine organisms have enabled an assessment of lethal thresholds (Vaquer-Sunyer and
Duarte, 2008; Rosewarne et al., 2016). These data coupled with recent advances in a theoretical
framework enable both explanatory and predictive power in the context of a dynamic
environment (Deutsch et al., 2015; Penn et al., 2018; Howard et al., 2020). The fundamental
insights here are that basal metabolic rates for ectothermic marine organisms depend on ambient
temperature and generally increase with warming (Gillooly et al., 2001). Increasing basal
metabolic rates impose additional demand for oxygen. Organisms use oxygen dissolved in
seawater and acquisition tends to be limited by diffusive processes; thus, oxygen supply is
related to the ambient $pO_2$. The ratio of oxygen supply to temperature-dependent demand
provides a critical indicator of the capacity for an organism to meet its metabolic requirements.
Deutsch et al. (2015) formalized these concepts into a quantity termed the "Metabolic Index
($\Phi$)", which is defined as the ratio of oxygen supply to an organism's resting metabolic demand.
Oxygen supply is parameterized according to a biomass-dependent scaling of $pO_2$, capturing
variation in the efficiency with which organisms acquire and utilize $O_2$. This can be expressed as
$S = \alpha_s B^{\sigma} pO_2$, where $\alpha_S$ is a mass-normalized coefficient expressing the rate of gas transfer
between an organism and its environment and $B^{\delta}$ is the scaling of supply with biomass, $B$ (Piiper
et al., 1971). Resting metabolic demand can be expressed using the Arrhenius equation as

$$D = \alpha_D B^{\delta} exp\{\frac{-E_d}{K_B}\left[\frac{1}{T} - \frac{1}{T_{ref}}\right]\},$$

where $\alpha_D$ is a species-specific basal metabolic rate, $B^{\delta}$ is the scaling of this rate with biomass, $E_d$
(eV) is the temperature dependence of oxygen supply, T is temperature, $T_{ref}$ is the reference
temperature (15°C), and $k_B$ is the Boltzmann constant (Gillooly et al., 2001). Gas transfer is
kinematically slow at low temperatures, and hence organism viability can be limited by the
energy to acquire oxygen at low temperatures, thus $E_o$ varies with temperature. Here we account
for this by adding the temperature dependence ($dE_o/dT$) to $E_o$ in equations above ($E_o + \frac{dE_o}{dT}(T -$
$T_{ref})$ ), using the mean value of $dE_o/dT = 0.022$ eV consistent with Deutsch et al. (2020). The
Metabolic Index can thus be written as the ratio of *S/D*:

$$\Phi = A_o pO_2 exp\{\frac{-E_o}{K_B}\left[\frac{1}{T} - \frac{1}{T_{ref}}\right]\}, \hspace{3cm} (1)$$





where $A_o = \alpha_S/\alpha_D$ *(1/atm)* is the hypoxic tolerance, $E_o = E_d - E_s$ ($E_s$ is the temperature
dependence of oxygen supply) (Deutsch et al., 2015; Penn et al., 2018). The exponent, $\varepsilon = \sigma -$
$\delta$, is the allometric scaling of the supply to demand ratio with biomass, is typically near zero.
Therefore, in the analysis that follows, we presume unit biomass and thus neglect potential
impacts of variations in biomass.

If $\Phi$ falls below a critical threshold value of 1, conditions are physiologically unsustainable: an
organism cannot meet its basic resting metabolic oxygen requirements. Conversely, values of $\Phi$
above 1 enable organismal metabolic rates to increase by a factor of $\Phi$ above resting levels,
permitting critical activities such as feeding, defence, growth, and reproduction. Thus, for a
given environment and species, $\Phi$ provides an estimate of the ratio of maximum sustainable
metabolic rate to the minimum rate necessary for basal metabolism. Deutsch et al. (2015)
inferred the ratio of active to resting energetic demand by examining the biogeographic
distribution of several species, finding that range boundaries coincide with values of $\Phi = 1.5–7$.
This threshold, termed critical rate ($\Phi_{crit}$), represents the minimum metabolic index required for
an organism to sustain an active metabolic state, which is a more meaningful ecological
threshold than requirements for resting metabolism. Therefore, in this study, we define a quantity
$\Phi'$, which is derived by dividing $\Phi$ by $\Phi_{crit}$; equivalently, this yields an adjusted definition of the
hypoxic tolerance trait, $A_c = A_o / \Phi_{crit}$, where $A_c$ is termed the "ecological hypoxia tolerance"
consistent with Howard et al. (2020). Where $\Phi' > 1$ (i.e., $\Phi > \Phi_{crit}$) an organism can sustain an
active metabolic rate; where $\Phi' < 1$ (i.e., $\Phi < \Phi_{crit}$), $O_2$ is insufficient and an active metabolic
state is not viable. Henceforth, our analysis uses $\Phi'$ to characterize ecotypes viability.

**2.2 Physiological dataset**
We make use of a dataset describing physiological parameters for a collection of 61 marine
ecotypes spanning a range of ecological hypoxic tolerances ($A_c$) and temperature sensitivities
($E_o$) (Penn et al., 2018; Deutsch et al., 2020, Figure 1a). We illustrate how the physiological traits
$E_o$ and $A_c$ constrain habitat viability in the context of distributions of $pO_2$ and temperature in the
marine environment in Figure 1b, which shows the minimum $pO_2$ (i.e., $pO_2$ at $\Phi_{crit}$) required to
sustain an active metabolic state as a function of temperature for five combinations of $E_o$ and $A_c$.



The five combinations are derived from sampling the probability distributions of $E_o$ and $A_c$
(Figure 1a) at the 10th, 50th, and 90th percentile values (illustrated by colored stars in Figure 1a
and corresponding curves in Figure 1b). We assume that the trait distributions are independent,
which is a reasonably modest simplification; $E_o$ is represented by a normal distribution and $A_c$ by
a lognormal distribution function (Figure S1). The $pO_2$ at $\Phi_{crit}$ curves shown in Figure 1b
delineate regions of $pO_2$-temperature space that are habitable (above the curve) and
uninhabitable (below the curve). The reversing curvature of $pO_2$ at $\Phi_{crit}$ in Figure 1b at low
temperature captures the decrease of the organism's oxygen acquisition efficiency in cooler
conditions yielding cold intolerance.

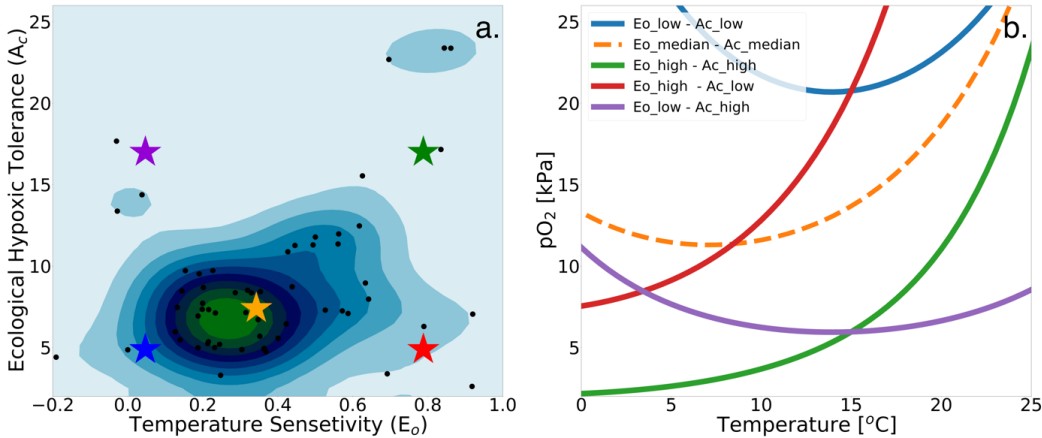


**Figure 1**. Physiological traits determining hypoxic tolerance. (a) Scatter plot of 61 marine ecotypes for which
empirically derived estimates of activation energy ($E_o$) and the ecological hypoxic tolerance ($A_c$) have been
determined (Penn et al., 2018). The color shows the density of occurrence for the 61 marine ecotypes in the $A_c$ - $E_o$
trait space. (b) The minimum $pO_2$ required to sustain an active metabolic state (i.e., $pO_2$ at $\Phi_{crit}$, Deutsch et al., 2020)
for five combinations of $A_c$ and $E_o$ corresponding to the stars in panel "a"; these are combinations of the 10th, 50th,
90th percentile values for each parameter.

To illustrate how the trait combinations of $E_o$ and $A_c$ exert control on the geographic distribution
of organisms in the marine environment (Deutsch et al., 2020), we use observations of $pO_2$ and T
along a zonal transect of the Pacific Ocean and plot $\Phi'$ for nine combinations of $E_o$ and $A_c$
percentile values (Figure 2). The colorbar in Figures 2a-i show the metabolic index for an active
state ($\Phi'$); regions with values above one are habitable (color), while regions with values below



one are uninhabitable (white) on the basis of metabolic constraints (other ecological
considerations are not considered). The subplots in the upper portion of the figure are arranged
according to the same trait axes shown in Figure 1a; $E_o$ increases horizontally from left to right
and $A_c$ increases from the bottom to the top. For the trait combination in the bottom left (low $E_o$,
low $A_c$; Figure 2g), metabolism is relatively insensitive to temperature, and tolerance for low
$p\mathrm{O}_2$ is poor. Thus, ecotypes with low $E_o$ and low $A_c$ are restricted to high latitude surface waters,
where temperatures are cool, and $p\mathrm{O}_2$ is abundant (Figure 2g). As $E_o$ increases from left to right,
metabolic rates become more sensitive to temperature. Then, habitat is gained at depth, where
temperatures are cooler and higher temperature sensitivity confers an advantage (Figure 2g–i).
From the bottom to the top, the increase in tolerance of low $p\mathrm{O}_2$ conditions increases habitability
in regions of low $p\mathrm{O}_2$, enabling organisms to expand beyond high-latitude surface waters (Figure
2g-a). The biogeographic range for organisms with high $A_c$ is modulated by $E_o$; as temperature
sensitivity increases, ecotype viability at high latitudes is increased, but tropical surface waters
become less viable (Figure 2 a-c). Henceforth, our analysis will utilize the metabolic index of the
median ecotype ($E_o = 0.34$, $A_c= 7.4$; Figure 2e) for illustrative purposes; i.e., all metabolic index
figures refer to this median ecotype unless otherwise stated.

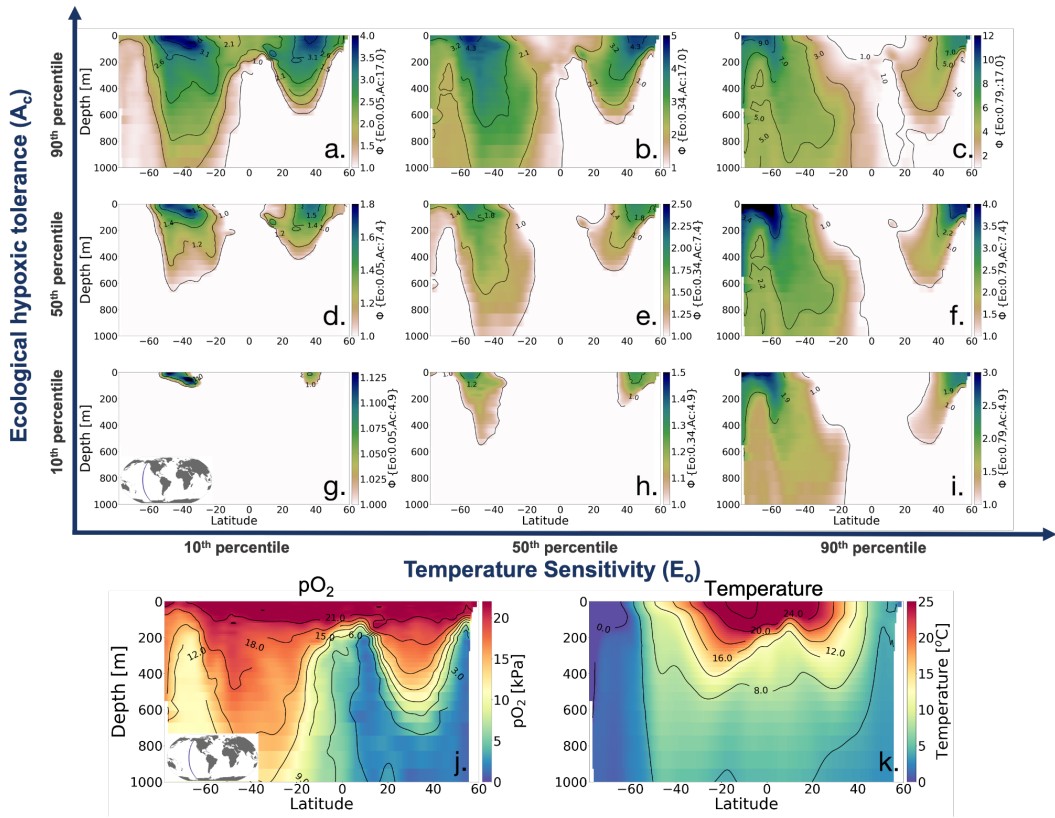

**Figure 2**. Annual mean metabolic index (Φ') for nine combinations of the ecological traits $E_o$ (metabolic temperature sensitivity) and $A_c$ (ecological hypoxic tolerance) along a transect in the Pacific Ocean based on a climatology from the World Ocean Atlas dataset (Garcia et al., 2014). The percentile values of each trait are: 10th ($E_o$ = 0.04, $A_c$= 4.8), 50th ($E_o$ = 0.34, $A_c$= 7.4), and 90th ($E_o$ = 0.79, $A_c$ = 17.0). The lower panels show $p$O$_2$ and temperature from the WOA dataset. Note that the colorbar range differs by panel and values where Φ' < 1 are omitted, thus the color shows only areas where an active metabolic state can be sustained.

### 2.3 Earth system model simulations

This study is based on the CESM1-LE, described in detail by Kay et al. (2015). The CESM1-LE included 34 ensemble members integrated from 1920–2100 under historical and RCP8.5 forcing. The ensemble was generated by adding round-off level ($10^{-14}$ K) perturbations to the air temperature field at initialization in 1920; this small difference yields rapidly diverging model solutions due to the chaotic dynamics intrinsic to the climate system, thus developing ensemble spread representative of internal variability (Kay et al., 2015). Briefly, the CESM1-LE uses the



Community Earth System Model, version 1 (Hurrell et al., 2013), with a horizontal resolution of
nominally 1° in all components. The ocean component is Parallel Ocean Program version 2,
(Smith et al., 2010) with sea ice simulated by the Los Alamos Sea Ice Model version 4 (Hunke
and Lipscomb, 2010). Ocean biogeochemistry was represented by the Biogeochemical Elemental
Cycling (BEC) model (Moore et al., 2013; Lindsay et al., 2014).

Our analysis focuses on three depths: 50 m representing near-surface dynamics, the epipelagic
zone at 200 m, and the mesopelagic zone at 500 m. $p$O$_2$ was calculated using the Garcia and
Gordon. (1992) solubility formulation. For convenience, we use the period 1920–1965 to define
a minimally-perturbed natural state, as this period is prior to the development of substantial
anthropogenic trends in ocean oxygen and temperature (Long et al., 2016). We also examine
distributions over the last three decades of the 21st century (2070–2099) to evaluate the projected
climate-change signal under RCP8.5. We use the mean across all 34 ensemble members to
quantify the deterministic, "forced" response of the climate system to anthropogenic influence
(Deser et al., 2012). The ensemble spread is thus indicative of the amplitude of variations
attributable to natural variability.



**Figure 3**. Mean-state comparison with observations. The climatological mean of (top rows) temperature (°C),
(middle rows) $pO_2$ (kPa), and the (bottom rows) metabolic index for active metabolism (Φ') for the median ecotype
($E_o$ = 0.34, $A_c$= 7.4); three depths are shown (left) 50 m, (center) 200 m, and (right) 500. Top panels show the
WOA13 dataset and the bottom panels show CESM1-LE.

We compared the CESM1-LE (1920 - 1965) with the World Ocean Atlas, version 2013
(WOA2013) dataset (Garcia et al., 2014), an observationally-based, gridded climatology (Figure
3a-i). CESM1-LE generally provides a reasonable representation of $pO_2$ and temperature



distributions at the selected depths (Figure 3); however, there are important biases to
acknowledge in the context of interpreting the results. Temperature magnitudes are generally
well simulated in the CESM1-LE, showing a root mean square error (RMSE) < 1.3 °C, and
pattern correlation coefficient (PCC) >0.98 in all three selected depths (50 m, 200 m, and 500)
(Table 1). Temperature magnitudes are slightly underestimated at 50 m and 200 m (mean bias of
< 0.3°C), and overestimated by 0.41 °C at 500 m. Note that since our comparison uses CESM1-
LE data from 1920-1965, some discrepancy in temperature might be expected from the signal of
climate warming present in the WOA observations. $pO_2$ is also reasonably well captured by the
CESM1-LE (PCC <0.95), but magnitudes are slightly underestimated at depth, showing a mean
bias of -1.63 kPa and -2.1 kPa at 200 m and 500 m with respect to WOA13 (Table 1). Regions of
low $pO_2$ waters are too extensive in CESM1-LE (Figure 3n-o) and there is a slight degradation of
skill with depth for $pO_2$ fields (Table 1). The underestimation of $pO_2$ leads to a slight
underestimation of Φ' with respect to WOA13 (Figure 3 p-r); however, Φ' computed from the
model fields demonstrates that the dominant spatial patterns are well captured by the CESM1-LE
despite magnitudes that are slightly too low (i.e., Figure 1, c, l). These differences ultimately
matter most near the hypoxic zones and at the boundaries of habitable zones like the OMZs.




**Table 1**. Summary statistics for the comparison of CESM1-LE with the World Ocean Atlas dataset (Garcia et al.,
2014). The columns include the mean bias, pattern correlation coefficient (PCC), and root mean square error
(RMSE) at 50 m, 200 m, and 500 m.

|  | **Mean bias** | **R** | **RMSE** |
|---|---|---|---|
|  | **Temperature [ºC]** | | |
| **50 m** | -0.17 | 0.99 | 1.22 |
| **200 m** | -0.25 | 0.99 | 1.22 |
| **500 m** | 0.10 | 0.98 | 0.63 |
|  | **pO$_2$ [kPa]** | | |
| **50 m** | 0.05 | 0.99 | 1.91 |
| **200 m** | -1.17 | 0.96 | 5.96 |
| **500 m** | -1.46 | 0.95 | 6.28 |
|  | **Metabolic index** | | |
| **50 m** | 0.01 | 0.99 | 0.02 |
| **200 m** | -0.09 | 0.97 | 0.05 |
| **500 m** | -0.15 | 0.96 | 0.08 |





## 2. Results

### 3.1 Joint temperature-$p$O$_2$ natural variability and forced trends

The spatial distribution of the number of viable ecotypes is shown in Figure 4 for the
"unperturbed" climate (1920-1965). Our intention here is not to quantify the actual
biogeographic range of organisms in the environment, but rather to illustrate the ocean's ability
to support respiration by marine ectotherms given the metabolic capacities afforded within the
trait space of extant organisms. High latitude environments do not impose strong aerobic
constraints (cold intolerance notwithstanding), thus over much of the Southern Ocean, North
Atlantic, and Arctic Ocean almost all 61 ecotypes can sustain respiration. The tropical oceans
impose the strongest aerobic constraints, restricting the viability of ecotypes that do not have
high-hypoxia tolerance ($A_o$). For example, less than 25 ecotypes are viable over much of the
tropical surface ocean (Figure 4a); low concentrations of oxygen at depth impose even stronger
constraints, and no ecotypes are viable in the core of OMZs (Figure 4b, c). The spatial patterns of
the number of viable ecotypes is tightly controlled by temperature at the surface, since $p$O$_2$ is
mostly near saturated levels; at depth, however, $p$O$_2$ is the dominant driver of geographic
patterns in ecotype viability (Figures 2-4). Temperature generally decreases with depth, reducing
the metabolic oxygen demand. However, since $p$O$_2$ also decreases with depth and displays
greater lateral heterogeneity, $p$O$_2$ emerges as the dominant constraint of spatial structure in
ecotype viability at depth.

The standard deviation of annual anomalies using all CESM1-LE ensemble members provides
insight into the amplitude of natural variability (Figure. 5). Temperature and $p$O$_2$ show similar
patterns of natural variability in the upper ocean, both showing particularly large variance in the
western tropical Pacific and Indian Ocean (Figure 5 a, d). Spatial variation in the magnitude of
temperature variability generally decreases with depth, but $p$O$_2$ displays even relatively larger
variability at depth with respect to the surface in some regions (Figure 5 a–f). The joint $p$O$_2$-
temperature variability manifests in variations of Φ' (Figure 5g-i). Natural variability in Φ'
computed for the median ecotype shows spatial patterns similar to temperature in the upper-





surface ocean (50 m), but is more similar to $pO_2$ at depth. Thus, variations in Φ' tend to be
temperature-dominated near the surface, but are more strongly controlled by $pO_2$ variability at
depth. Φ' also shows the most extensive natural variability at 200 m consistent with the
variability of $pO_2$. The number of viable species shows more dramatic fluctuations than
variations in the median ecotype Φ'; variations in the number of viable ecotypes exceed 30% on
annual timescales in the tropical upper ocean and near OMZ boundaries in the water column
(Figure 4 c–d). This reflects the fact that interannual variability can preclude habitability for
some regions of the $A_c$-$E_o$ trait space, but these variations do not necessarily impact viability for
the median ecotype (Figure 1). In the tropical surface ocean, high temperatures (>25°C), and
saturated surface ($pO_2$ >20 kPa) require high hypoxia tolerance ($A_c$), but permit a range of
$E_o$ values (Figure 1b, 2a-b). Ecotypes with larger temperature sensitivity (high $E_o$) are
particularly responsive to variations in temperature.

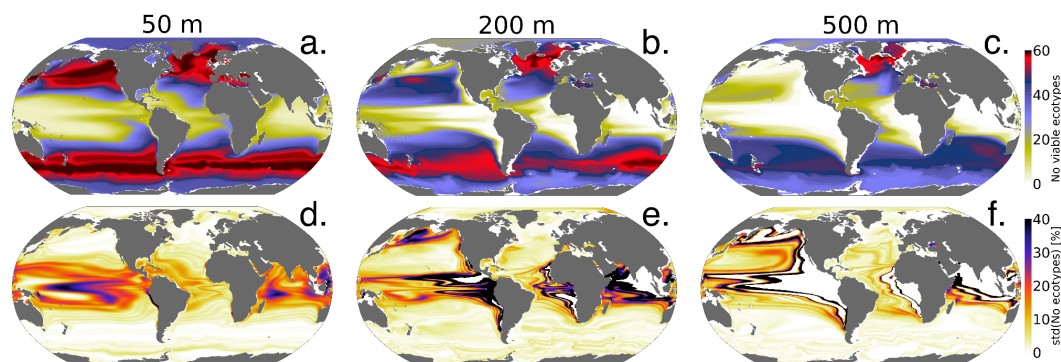


**Figure 4**. Metabolic constraints on trait-space viability. Top row: the number of ecotypes from the physiological
trait database that are viable (total = 61) in the CESM1-LE over the period 1920–1965. Bottom row: the standard
deviation (expressed as a percent of the mean) in the number of viable ecotypes, reflecting fluctuations driven by
natural variability.

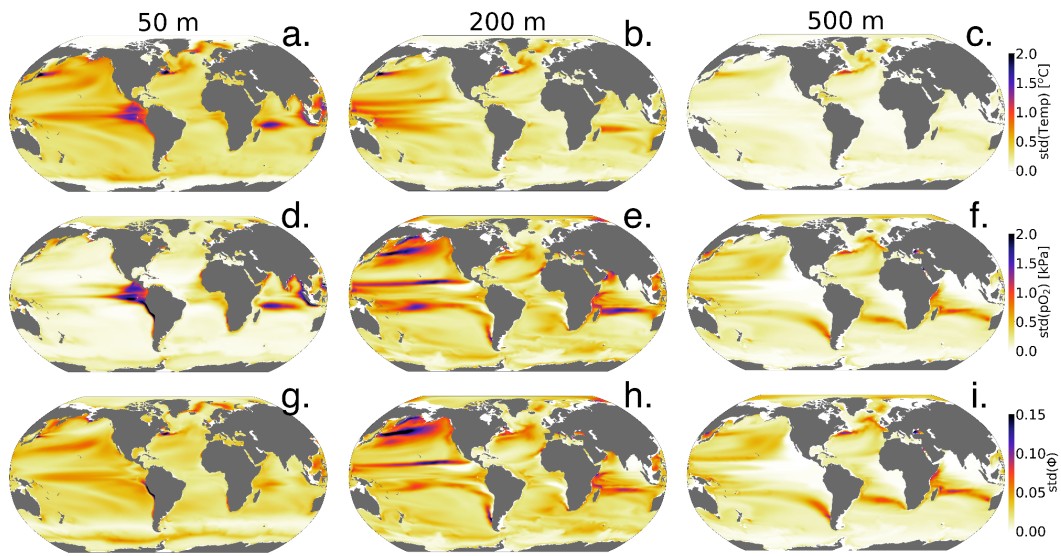


**Figure 5**. The amplitude of natural variability in the ocean's metabolic state. The panels show the standard deviation
of annual-mean anomalies of all ensemble members over the period 1920–1965 for (top row) temperature (ºC),
(middle row) $p$O$_2$ (kPa), and (bottom row) the metabolic index (unitless) of the median ecotype ($E_o$ = 0.34, $A_c$= 7.4).

CESM1-LE simulates nearly homogeneous warming between 1920–1965 and 2070–2099 in the
surface ocean (50 m) under RCP8.5, with an exception of the so-called North Atlantic warming
hole (Figure 6a). Both modelling and observational studies have linked the North Atlantic
warming hole to the slowing of the Atlantic overturning circulation with climate change  (Keil et
al., 2020). The magnitude of ocean warming generally diminishes with depth except in the North
Atlantic, where, despite reductions, the overturning circulation effectively propagates
anthropogenic heat anomalies into the ocean interior. $p$O$_2$ shows heterogeneous changes between
1920–1965 and 2070–2099 (Figure 6 d-f). In the upper ocean, $p$O$_2$ changes are generally small (<
1 kPa) because the near-surface is kept close to saturation via photosynthetic oxygen production
and air-sea equilibration. At depth, however, $p$O$_2$ shows long-term changes linked to
accumulated effects of respiration and changes in circulation (Ito et al., 2017). At 200 m for
example, the Pacific Ocean displays a basin-wide mean reduction in $p$O$_2$ of 2 kPa (~30%), while
the Atlantic and Indian basins gain about >2 kPa (~ 10 - 35%) by the end of the century. The
largest long-term $p$O$_2$ loss (>3 kPa) occurs in the North Pacific while the largest $p$O$_2$ gain (~2



kPa) occurs in the North Atlantic gyre and western Indian Ocean (Figure 6 e-f).

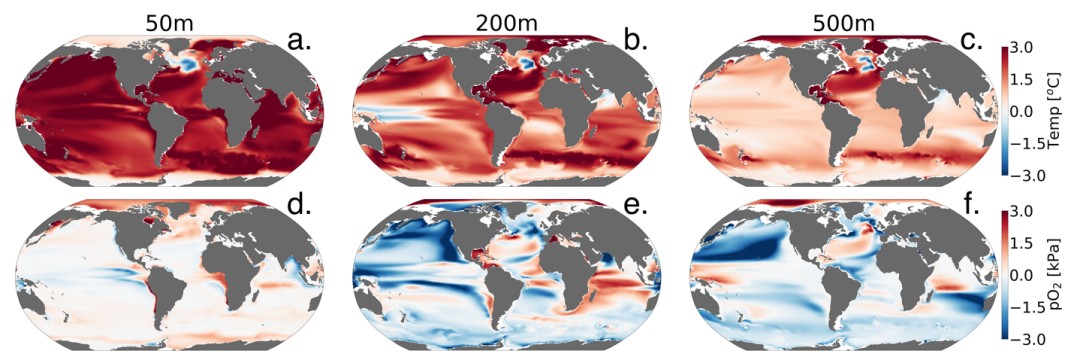


**Figure 6**. Net long-term change (2070–2099 minus 1920–1965) in the CESM1-LE ensemble means temperature
(top) and (bottom) $pO_2$ at 50 m, 200 m, and 500 m.

Figure 7 shows the relationship between interannual variations in $pO_2$ versus temperature ($pO_2$-
T) in the unperturbed climate (1920–1965; top row) and for the forced trend associated with 21st
century climate change (2070–2099 minus 1920–1965; bottom row). The nature of the $pO_2$-T
relationship is an important indicator of the impacts of variability on the metabolic state.
Furthermore, the extent to which the forced trend is characterized by a $pO_2$-T relationship that is
distinct from that associated with natural variability provides insight into the potential for
advanced or delayed detection of signals in Φ relative to $pO_2$ or temperature alone. Given that
metabolic rates for most organisms increase with temperature (positive $E_o$), a positive correlation
between variations in temperature and $pO_2$ is generally indicative of compensating changes,
wherein increased oxygen demand is at least partially offset by increased supply. Anticorrelation
between temperature and $pO_2$, by contrast, will generally be associated with compounding
impacts on the metabolic index, as a negative correlation indicates that reductions in $pO_2$ (i.e.,
oxygen supply) accompany warming (i.e., increased demand). The sign of the $pO_2$-T relationship
in the natural climate varies regionally and with depth (Figure 7, top row). The surface ocean is
generally characterized by a weak, positive $pO_2$-T relationship, which could manifest from,
among other mechanisms, temperature-induced increases in photosynthetic oxygen production
(Figure 7a). The natural $pO_2$-T relationship in the epipelagic (200 m) is characterized by strong
positive correlations in the tropics and negative correlations at high latitudes (Figure 7b). A
positive correlation between $pO_2$ and temperature at this depth could be induced by variability



associated with adiabatic vertical displacement of isopycnals, or "heave", which has the effect of
translating background gradients in properties vertically in the water column. Upward movement
of a deep isopycnal surface would yield a negative temperature anomaly and a negative $p$O$_2$
anomaly (positive correlation), as the deeper, colder waters have greater oxygen utilization
signatures associated with longer ventilation age. Negative correlations between $p$O$_2$ and
temperature could manifest from ventilation processes, where enhanced subduction of surface
water yields anomalously cold water masses that are enriched in oxygen. The sign of these
epipelagic $p$O$_2$-T correlations shows some similarity to those associated with the externally
forced climate (Figure 6e), but the latter is characterized by a greater prevalence of
anticorrelation, most notably in the North Pacific ocean. At 500 m depth, the relationship
between temperature and $p$O$_2$ in the natural climate is almost a mirror image of the epipelagic
(Figure 7c); the tropics generally display negative correlations, while polar regions show positive
correlations (Figure 7 e). The $p$O$_2$-T relationship in the forced trend at 500 m is dominated by
broad regions of deeply negative correlations, with the most pronounced effect again in the
North Pacific. The negative relationship is consistent with a ventilation signal, as buoyancy-
induced stratification from warming curtails the introduction of new oxygen into the ocean
interior. The predominantly negative $p$O$_2$-T relationship associated with the forced trend is
indicative of the compounding effects of climate change on metabolic state, increasing metabolic
demand while simultaneously reducing oxygen supply.

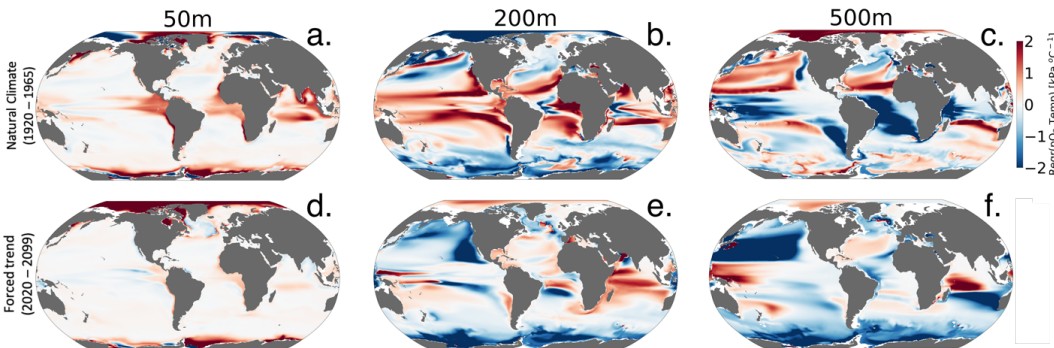


**Figure 7.** Regression of annual means $p$O$_2$ versus temperature (kPa °C$^{-1}$) for (top row) interannual variability and
(bottom row) the forced trend (difference between 2020–2099 and 1920–1965). The columns show the regressions
computed at different depths, 50 m, 200 m, and 500 m, respectively.



### 3.2 Long-term habitat changes

Figure 8 shows the climate-driven changes in Φ' for the median ecotype, as well as the impacts of climate change on the number of viable ecotypes. Notably, while $pO_2$ in the near-surface ocean is relatively insensitive to climate change (Figure 6d), there are reductions in Φ' in the tropics (Figure 9d), owing to the direct impacts of warming. These changes are associated with deep reductions in the number of viable ecotypes in the tropics (Figure 8a). There are modest increases in Φ' and ecotype viability at high-latitudes; metabolic state in these regions is affected by cold intolerance, thus warming broadens the viable region of trait space. Additionally, reductions in sea ice cause an increase in $pO_2$, as gas exchange becomes more effective at restoring equilibrium oxygen concentrations. The number of viable ecotypes shows more intense patterns than those in the median ecotype Φ' in the upper ocean (Figure 8). This is partly because ecotypes predicted to lose viability in the tropical regions (~ 50%) are at the extremes of the $A_c$-$E_o$ distribution (Figure 1) and not captured by the median ecotype Φ'. Nevertheless, outside the tropical regions, the median ecotype gives a good indication of the anthropogenic impact to marine ectotherms. The projected habitat loss in the epipelagic-pelagic North Pacific (> 50%) and habitat gain in the epipelagic-pelagic Southern Indian Ocean (~40%) and pelagic western tropical regions (~40%) are consistent with a decrease in the median ecotype Φ'. Note that the most pronounced effects on habitat are associated with regions where climate change drives a strongly negative $pO_2$-temperature relationship (Figure 7).

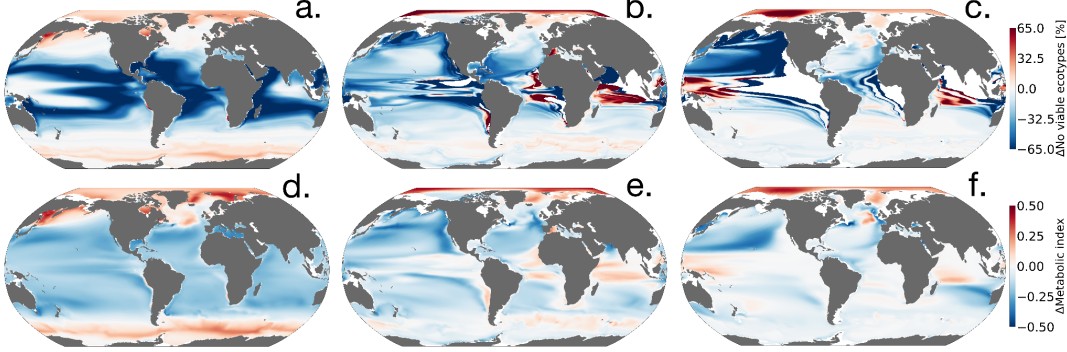

**Figure 8**. Net change in the number of habitable ecotypes in percentage (top row). Net metabolic index change (2070 - 2099 vs. 1920 - 1965) for the median ecotype [$E_o$ = 0.34, $A_c$= 7.4] (bottom row). At 50m (first column), 200m (second column) and 500m (third column).




**3.3 Time of Emergence**


In this section, we examine the "time of emergence" (ToE, Hawkins and Sutton, 2012), the point
when forced changes in $pO_2$, temperature and $\Phi'$ can be distinguished from the background
natural variability. We define ToE as the time when the magnitude of change in the ensemble
mean of a particular variable exceeds two standard deviations of the natural climate (1920 -
1965). This is illustrated in Figure 9 for a single grid point in the North Pacific at 200 m. At this
location, the forced trend in temperature shows a monotonic increase, while $pO_2$ shows a
monotonic decrease; as a result, $\Phi'$ for the median ecotype and the number of viable ecotypes
decrease over time. The anti-correlation between $pO_2$ and temperature exacerbates trends in $\Phi'$,
and hence the forced trend of the median ecotype $\Phi'$ emerges from natural noise earlier than
either $pO_2$ or temperature do alone (Figure 10a-c). Note that although the ToE of ecotype
viability change is directly derived from changes in $\Phi'$, it is binary counted; changes in ecotype
viability are counted in whole numbers and this creates a step-function temporal-spatial variation
(Figure 9d). Consequently, this step-function-like feature of ecotype viability creates
discontinuities even in spatial patterns of ToE (Figure 10 j-l) as also shown in the natural
variance in Figure 4 d-f.




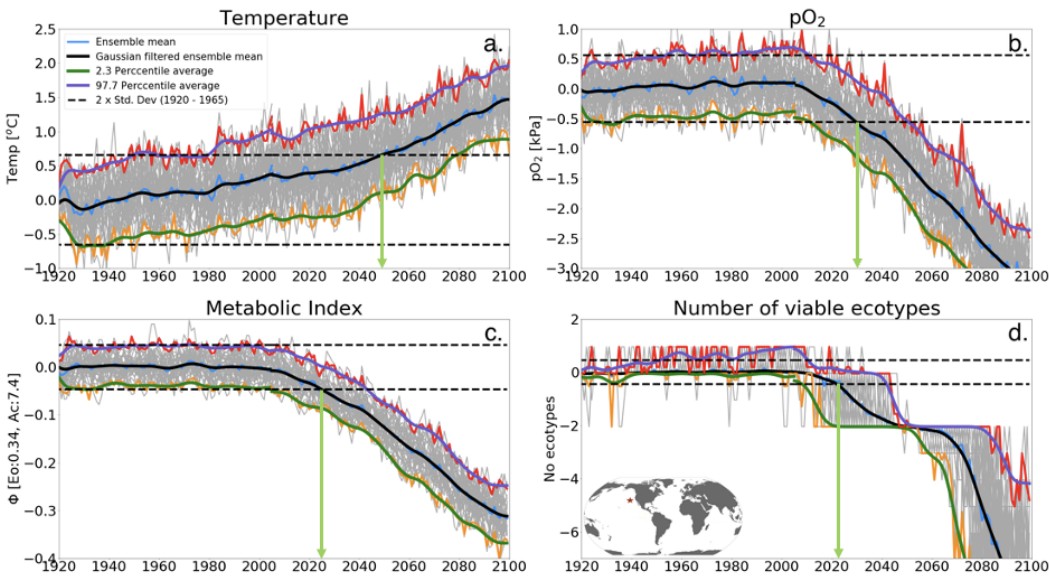

**Figure 9.**. Time of emergence (ToE) of the climate forcing signal for (a) temperature, (b) $pO_2$ (c) the metabolic
index of the median ecotype [$E_o = 0.34$, $A_c = 7.4$], and (d) the number of viable ecotypes for a single model grid in
the North Pacific at 200 m. ToE (green arrows) is defined as the time when the forced trend signal (ensemble
member time series) is above two standard deviations (black dotted line) of all ensemble members for the period
1920 - 1965.



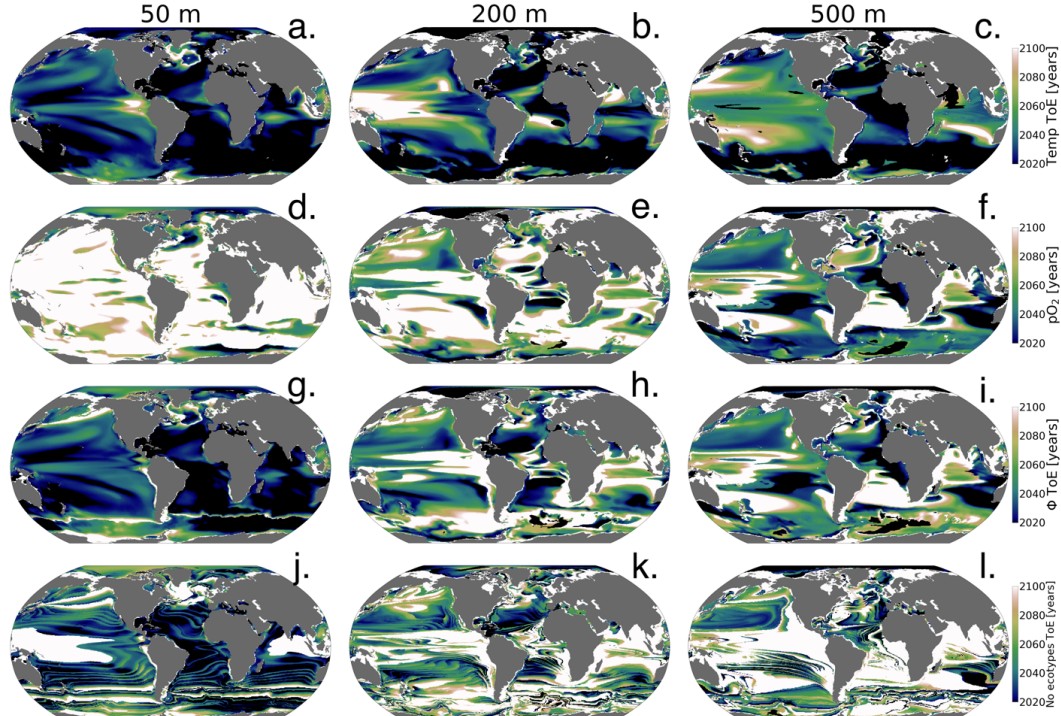

**Figure 10**. Time of emergence (ToE) of the climate forcing signal for temperature, $pO_2$, phi, and the number of viable ecotypes. ToE is defined as the time when the forced trend signal (ensemble member time series) is above two standard deviations of all ensemble members for the period 1920 - 1965.

The ToE of $pO_2$ and temperature are inverted with depth; temperature emerges earliest in the upper ocean while $pO_2$ emerges earlier at depth and later or shows no emergence in the upper ocean (Figure 10 a-f). This feature is consistent with larger upper ocean temperatures long-term changes and greater $pO_2$ changes at depth. Near-surface ocean temperature has mostly already emerged by 2020 and is predicted to have almost completely emerged by the late 2060s under RCP85 (Figure 10 a-c). The early emergence of temperature from natural noise also persists for regions of relatively low natural variance at depth, e.g., the Southern Ocean and Atlantic Basin Gyres. Regions of the largest natural variability (see Figure 5) like the subtropical-subpolar Pacific however do not emerge until close to the end of the century. For $pO_2$, anthropogenic changes in the upper ocean generally do not emerge from natural noise before the end of the century except for the Arctic Ocean and Eastern Antarctic. In the Arctic Ocean and Eastern





Antarctic $p$O$_2$ gain is related to sea-melt emergence by the mid-2050s (Figure 10a). The median
ecotype Φ' ToE shows spatial patterns that are coherent with temperature ToE in the upper ocean
with exception of polar regions. In contrast, they are consistent with pO$_2$ ToE patterns at depth;
this is consistent with net long-term Φ' changes in Figure 9d. The emergence of the
anthropogenic signal in ecotype viability closely resembles the median ecotype Φ' spatial
patterns but showing non-harmonious spatial patterns due to the step-function-like counting
feature of viability changes. It shows that the predicted ∼ 50% ecotype viability loss in the
tropics (Figure 6a) may already be distinguishable from natural variability by the mid-2030s. In
the North Pacific, the predicted > 50% ecotype viability loss in the epipelagic-pelagic regions is
predicted to start emerging in the 2040s at 500 m and 2080s at 200 m (Figure 10 k-l).

**4. Discussion**

The human-induced rapid warming of the planet has been shown to drive ocean deoxygenation
(Ito et al., 2017; Schmidtko et al., 2017; Long et al., 2016). Higher metabolic oxygen demand at
higher temperatures (Gillooly et al., 2001; Deutsch et al., 2015, 2022) raises concerns about the
ability of marine ectotherms to support aerobic respiration in the future. This study set out to
characterize the anticipated climate change signal in the ocean's metabolic state in the context of
natural variability using the metabolic theory as a basis to examine the capacity of the
environment to support ectothermic marine heterotrophs.

The spatial variation in $p$O$_2$ and temperature in the unperturbed natural climate state set
biogeographic boundaries based on ectotherms' physiological performance. The resilience of
these ectotherms' biogeographic structure to natural variability and long-term climate warming is
perturbed by the joint $p$O$_2$-temperature changes, effectively measured by the metabolic index
(Φ'). An increase in the capacity of the organisms to support aerobic respiration increases Φ'; for
example by ocean cooling or increase in oxygen supply contrary, warming and decrease in
oxygen supply decrease Φ'. There are exceptions in extremely low-temperature environments,
where aerobic respiration is also limited by kinematic gas transfer into the organism in addition
to environmental oxygen supply. Relative changes in $p$O$_2$ and temperature in the natural



variability and forced trend, therefore, regulate ectotherms' resilience to environmental changes.
Under the RCP85 climate scenario, the ocean generally warms homogeneously but concurrent
$pO_2$ changes are heterogeneous and vary with depth. Thus, the characteristics of these $pO_2$-
temperature forced trend changes determine when the climate change impact on marine
ectotherms can be distinguishable from natural variability.

In the surface ocean, $pO_2$ is generally abundant and relatively uniform, and thus spatial
temperature variations have a dominant constraint on the spatial variations of organismic
metabolic state. The warmest parts of the surface ocean, the tropical oceans, can only support
about 10-20 ($\sim$ 30%) of the 61 ecotypes while cooler regions in extratropics have nearly 100%
viability. Moreover, since warming anomalies propagate from the surface, the surface tropical
oceans also show the largest natural variance in temperature and ecotype viability in the surface
ocean. This is because extremely warm temperatures in the surface tropics (>25ºC) are mainly
suited for organisms with high-temperature sensitivity ($E_o$), which are relatively fewer, and
mostly close to their physiological limits (Storch et al., 2014). Large natural variability in these
warmest parts of the tropical surface ocean precludes the forced trend signal from emerging from
the natural variability in the ecotype viability by end of the century although the ocean warms the
largest in the surface. Nevertheless, the large warming trends in the surface ocean generally
emerge relatively early (the 2020s) from natural variability in both temperature and ecotype
viability in most regions. Minimal changes in surface $pO_2$ in the forced trend affirm that surface
ocean marine ectotherms are mainly perturbed by temperature in the context of anthropogenic
changes. In polar regions, warming has a counterintuitive effect on marine ectotherms with
respect to most parts of the surface ocean. There, warming helps organisms escape extreme cold
intolerances by enhancing membrane kinematic gas transfer which enhances $\Phi'$ and thus ecotype
richness in the future.

In the epipelagic and pelagic regions (200 m and 500 m), the temperature forced trend and
natural variability are smaller compared to the surface ocean, while concurrent $pO_2$ changes are
larger than the surface ocean. Thus, $pO_2$ and temperature play a more intricate role in
perturbating marine ectotherm habitats in the context of anthropogenic warming with respect to
the surface ocean, where temperature plays a dominant role. At depth, contrasting the regression



between $p$O$_2$ and temperature in the natural climate, and forced trends provides an instructive
framework to analyzing ectotherms' long-term changes. Regions showing distinct correlations
between temperature and $p$O$_2$ in the forced trends relative to the natural variability show a
weakening metabolic resilience; loss of habitat and emerging relatively early from natural
variability. For example, in the pelagic - epipelagic North Pacific, temperature-$p$O$_2$ regressions
switched from a positive correlation in the unperturbed climate to a strong negative correlation in
the forced trend. Consequently, the pelagic-epipelagic North Pacific is projected to lose nearly
half of the present climate ecotype viability by the end of the century. This loss of pelagic -
epipelagic North Pacific habitat is projected to emerge earliest at 500 m (the 2030s) where
anthropogenic $p$O$_2$ losses are larger than at 200 m. On the other hand, in the Arctic Ocean and
some parts of the Southern Ocean, concomitant $p$O$_2$-temperature correlations in the forced trends
result in the preservation of the marine habitat and even slight enhancements.

**5. Conclusions**

The joint temperature-oxygen metabolic framework in this study provides additional insight into
the impact of climate change on marine ecosystems in comparison to the independent oxygen or
temperature analysis. We here showed that while warming is the leading order driving
mechanism of climate change, the direct effect of warming on marine ecosystems is mostly in
the upper ocean. Climate change-related oxygen loss is a major driver of marine ecosystem stress
in addition to warming at depth. Incorporating organismal physiological sensitivity to oxygen-
temperature changes in the metabolic framework provides insight into how climate impacts the
biogeographic structure of marine habitat. We find that underway forced trends perturbations in
$p$O$_2$ and temperature will strongly exceed those associated with the natural system in many parts
of the upper ocean, mostly pushing organisms in these environments closer to or beyond their
physiological limits. Climate warming is expected to drive significant marine habitat loss in the
surface tropical oceans and epipelagic - pelagic North Pacific Basin, while gaining marginal
habitat viability in the surface Arctic Ocean and some parts of the Ocean Southern.

**6. Competing interests**



The contact author has declared that none of the authors has any competing interests

## 7. Acknowledgments


PM, ML, CD and TI were funded by the National Science Foundation (NSF) grant agreement
No. 1737158. PM and YSF were also funded by the European Union's Horizon 2020 research
and innovation programme under grant agreement No. 820989 (COMFORT). ). We also would
like to acknowledge the data access and computing support provided by the NCAR Cheyenne
HPC.

## 8. Author contribution


PM and ML designed the study approach. PM developed the model code and analysis with
feedback from ML, CD and TI. PM prepared the manuscript with contributions from all co-
authors.

## 9. Data access


The CESM1 large ensemble data used in this study can be accessed in this location:
https://www.cesm.ucar.edu/community-projects/lens/data-sets

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
