# Peer review of "Climatic Controls on Metabolic Constraints in the Ocean"

_EGUsphere, 2023_

## Referee Comment (RC1)

**Title**: Climatic Controls on Metabolic Constraints in the Ocean
**Lead Author**: Precious Mongwe
**Journal**: EGUsphere
**Date**: 3/2/2024

**Summary:**
This paper is quite interesting and logically organized. The motivating questions are made clear in the Introduction and the figures are appropriately used to tell the story. In general, the writing is quite clear outside of the Methods section, the last paragraph of the Discussion, and portions of the Conclusions paragraph. The authors will need to correct what seem to be multiple typos throughout the Methods section before this can be published, so I am **recommending minor revisions**. I also recommend that the authors consider adding a schematic to visually clarify relationships between key metrics of the paper. This would increase the accessibility of the paper significantly and serve as a valuable reference for the Discussion section, particularly when explaining some of the more complex impacts of oxygen and temperature changes on ectotherm habitability of high and low latitudes.

**Comments and Suggested Edits:**
Line 27: Typo. Signals → signal

Line 44: It may be valuable to incorporate the concept of higher oxygen demand, independent of oxygen supply or circulation changes.

Lines 109-110: Typo? $B^\sigma$ is in the equation but you define $B^\delta$

Line 117: $E_0$ is undefined at this point, so this section of text is confusing.

Equation 1: The B term is missing, and this equation should be labeled equation 2. Even though the B term is ultimately dropped, it should be included for clarity, following Deutsch et al., 2020 equation 1.

Line 123: the $E_S$ term was not included in the supply equation, should it be?

There are a lot of equations buried in the text – I would advise pulling them out and double checking all notations for consistency.

The text describing $\Phi_{crit}$ is confusing. Is the goal to say that the minimum $\Phi$ value found within the habitat domain of a given species, based on available data, reflects an empirical estimate of $\Phi_{crit}$ value? If so, please state more clearly.

Line 139: This text is not clear. Please stick to one concept at a time. For example: "$\Phi'$ is derived by dividing $\Phi$ by $\Phi_{crit}$, so when $\Phi$ falls below 1, the organism can no longer sustain its active metabolic demand and will need to make physiological tradeoffs. Account for these active metabolic requirements, we use an adjusted definition of the hypoxic tolerance trait, $A_c = A_o / \Phi_{crit}$, where $A_c$ is termed the "ecological hypoxia tolerance", consistent with Howard et al., 2020."

Line 161: It's not clear how this relationship yields cold tolerance, please elaborate, or reword for accuracy.

Figure 1b. It may help to clarify in the figure caption that below the $pO_2$ lines shown, the organism would experience an oxygen deficit relative to its active metabolism requirements, effectively signifying the species-specific hypoxic conditions, based on physiological traits, for this range of temperatures.

Figure 2: Center the global map on the Pacific to make the transect location easier to see.

Figure 3: Add prime to Φ color bars.

Line 347: Can you validate this hypothesis by looking at interannual variations in model density versus temperature or oxygen?

Figure 8 Caption: Note that the same decades for differencing apply to the top row of plots in addition to the bottom row.

Line 480: Is this a typo? Aren't high temperature regions mostly suited for organisms with high-temperature tolerance or reduced temperature sensitivity (Figure 2)?

Line 494: Should this say epipelagic and **meso**pelagic? This entire paragraph stands out as being particularly unclear relative to all other text (outside of the methods).

Line 498: Sentence starting with "At depth" could be reworded for clarity.

Line 500: By "distinct" do you mean correlations of opposite sign?

Line 509: It's not clear what is meant by "concomitant $p$O$_2$-temperature correlations in the forced trends". I assume this means trends of the same sign, but it would be ideal if this were clearly stated.

Line 516: Suggest changing to: "We here showed that, while warming is the primary mechanism driving climate change, the direct effect of warming on marine ecosystems is largely confined to the upper ocean."

Line 521: Suggest changing to: "We find that forced perturbations to $p$O$_2$ and temperature will strongly exceed those associated with the natural system…"

---

## Author Response (AR1)

**Rebuttal report**

**Climatic Controls on Metabolic Constraints in the Ocean**

Precious Mongwe[1], Matthew Long[2], Takamitsu Ito[3,] Curtis Deutsch[4], and Yeray Santana-Falcón[5]

[1]Southern Ocean Carbon Climate Observatory (SOCCO), CSIR, Cape Town, South Africa

[2]Oceanography Section, Climate and Global Dynamics Laboratory, National Center for Atmospheric Research, Boulder, CO, United States of America

[3]School of Earth and Atmospheric Sciences, Georgia Institute of Technology, Atlanta, Georgia  United States of America

[4]Department of Geosciences, Princeton University, Princeton, NJ, United States of America

[5]CNRM, Université de Toulouse, Météo-France, CNRS, Toulouse, 31057, France

**Reviewer1**

This paper is quite interesting and logically organized. The motivating questions are made clear in the Introduction and the figures are appropriately used to tell the story. In general, the writing is quite clear outside of the Methods section, the last paragraph of the Discussion, and portions of the Conclusions paragraph. The authors will need to correct what seem to be multiple typos throughout the Methods section before this can be published, so I am recommending minor revisions. I also recommend that the authors consider adding a schematic to visually clarify relationships between key metrics of the paper. This would increase the accessibility of the paper significantly and serve as a valuable reference for the Discussion section, particularly when explaining some of the more complex impacts of oxygen and temperature changes on ectotherm habitability of high and low latitudes.

**Response**: We thank the reviewer for his/her insightful comments and suggestions. Indeed, the methods section had multiple typos in the equations, we apologies for this. We have addressed these typos as pointed out, and we will consider adding a schematic diagram to tie together and summarize the paper.

Line 27: Typo. Signals → signal

**Response**: Addressed as suggested

Line 44: It may be valuable to incorporate the concept of higher oxygen demand, independent of oxygen supply or circulation changes.

**Response:** We would like to address this comment; however, it is unclear what the reviewer is suggesting or pointing out with reference to line 44. There may be a mistake in the line reference.

Lines 109-110: Typo? $B$ σ is in the equation but you define $B\delta$

**Response**: We thank the reviewer for point this out, this was indeed a typo, it is now corrected.

Equation 1: The B term is missing, and this equation should be labeled equation 2. Even though the B term is ultimately dropped, it should be included for clarity, following Deutsch et al., 2020 equation 1.

**Response**:  The B term was indeed excluded because it drops out, we have now included the full form of equations (Line 112 – 126).

Line 139: This text is not clear. Please stick to one concept at a time. For example: "Φ' is derived by
dividing Φ by Φcrit , so when Φ falls below 1, the organism can no longer sustain its active metabolic
demand and will need to make physiological trade-offs. Account for these active metabolic
requirements, we use an adjusted definition of the hypoxic tolerance trait, A c = A o / Φcrit , where A
c is termed the "ecological hypoxia tolerance", consistent with Howard et al., 2020."

**Response**: We thank the review for this suggestion, it was implemented as suggested, Line 143 – 150

***Line 143 – 150*** "Therefore, in this study, we define a quantity  Φ' derived by dividing Φ by $\Phi_{crit}$, so
when Φ falls below 1, the organism can no longer sustain its active metabolic demand and will need
to make physiological trade-offs. Account for these active metabolic requirements, we use an adjusted
definition of the hypoxic tolerance trait, $A_c = A_o / \Phi_{crit}$, where $A_c$ is termed the "ecological hypoxia
tolerance", consistent with Howard et al., 2020. Where Φ' > 1 (i.e.,  $\Phi > \Phi_{crit}$) an organism can sustain
an active metabolic rate; where Φ' < 1 (i.e., $\Phi < \Phi_{crit}$), $O_2$ is insufficient and an active metabolic state
is not viable. Henceforth, our analysis focuses on Φ'; in the subsequent Φ' = Φ for the text and
figures."

Line 161: It's not clear how this relationship yields cold tolerance, please elaborate, or reword for
accuracy.

**Response**: This is illustrated in Fig. 1b, where the nearly parabolic curvature of $pO_2$ at Φcrit indicates
an increase in oxygen demand at both low temperatures and high temperatures. Most of the
manuscript focuses on the high-temperature oxygen demand based on metabolic demand.
Nevertheless, at very low temperatures, gas transfer is limited by the decrease in molecular gas
diffusion, and as a consequence, oxygen transfer into the organisms requires energy, leading to cold
intolerance. We extend the text make the discrepcion clearer.
***Line 169 – 173*** "The reversing curvature of $pO_2$ at $\Phi_{crit}$ in Figure 1b at low temperature captures the
decrease of the organism's oxygen acquisition efficiency in cooler conditions yielding cold
intolerance. At very low temperatures, gas transfer is limited by the decrease in molecular gas
diffusion, as a consequence, oxygen transfer into the organisms requires energy, yielding cold
intolerance, this is well illustrating by the blue line in Figure 1b."

Figure 1b. It may help to clarify in the figure caption that below the pO2 lines shown, the organism
would experience an oxygen deficit relative to its active metabolism requirements, effectively
signifying the species-specific hypoxic conditions, based on physiological traits, for this range of
temperatures. Figure 2: Center the global map on the Pacific to make the transect location easier to
see.

**Response**: We thank the reviewer for this suggestion, we added the suggested description in Figure
1b

Figure 3: Add prime to Φ color bars.

**Response**: We have now add general comment in methods to clarify that Φ text refers to Φ'
throughout the text according to Howard et al., 2020's definition.
**Line 148 – 150**: "Where Φ' > 1 (i.e.,  $\Phi > \Phi_{crit}$) an organism can sustain an active metabolic rate;
where Φ' < 1 (i.e., $\Phi < \Phi_{crit}$), $O_2$ is insufficient and an active metabolic state is not viable. Henceforth,
our analysis focuses on Φ'; in the subsequent Φ' = Φ for the text and figures."

Line 347: Can you validate this hypothesis by looking at interannual variations in model density versus temperature or oxygen?

**Response**: We thank the reviewer for this comment, we have referenced Long et al., 2016 where this hypothesis is discussed.

Figure 8 Caption: Note that the same decades for differencing apply to the top row of plots in addition to the bottom row.

**Response**: We added a title to figure 8

Line 480: Is this a typo? Aren't high temperature regions mostly suited for organisms with high-temperature tolerance or reduced temperature sensitivity (Figure 2)?

**Response**: This is not a typo; this phenomenon is better explained by Figure 1.b. Due to high temperatures in the tropics, habitability requires either high oxygen tolerance or high temperature sensitivity (high $E_o$). High $E_o$ organisms have particularly strong temperature sensitivity at high temperatures.

Line 494: Should this say epipelagic and mesopelagic? This entire paragraph stands out as being particularly unclear relative to all other text (outside of the methods).

**Response:** Indeed, this was a typo and we apologies for this sloppy paragraph. We have updated this paragraph

**Line 529 – 544**: "In the epipelagic and mesopelagic regions (200 m and 500 m), the forced temperature trend and natural variability are broadly smaller than the surface ocean, while $pO_2$ changes show the opposite. Thus, at depth $pO_2$ play a more intricate role in perturbating marine ectotherm habitats in the context of anthropogenic warming with respect to the surface ocean, where temperature plays a dominant role. Contrasting the regression between $pO_2$ and temperature in the natural climate, and forced trends provides an instructive framework to analysing ectotherms' long-term changes. Regions showing different correlations between temperature and $pO_2$ in the forced trends in comparison to the natural climate suggest a loss metabolic resilience; loss of habitat, and these regions tend to have a relatively early ToE. For instance, in the epipelagic and mesopelagic North Pacific, temperature-$pO_2$ regressions switched from a positive correlation in the unperturbed climate to a strong negative correlation in the forced trend (Figure 7). The North Pacific pelagic – epipelagic regions is projected to lose nearly half of the present climate ecotype viability by end of the $21^{st}$ century, the projected habitat loss start emerging by the late 2030s under the RCP85 climate scenario, On the other hand, in the Arctic Ocean and some parts of the Southern Ocean, same sign $pO_2$-temperature correlations in the forced trends result in the preservation of the marine habitat and even slight enhancements."

Line 498: Sentence starting with "At depth" could be reworded for clarity.

**Response**: This entire paragraph is reformulated.

Line 500: By "distinct" do you mean correlations of opposite sign?
**Response**: Distinct is replaced by "differences" which clarify the meaning of the sentence.

Line 509: It's not clear what is meant by "concomitant pO2 -temperature correlations in the forced
trends". I assume this means trends of the same sign, but it would be ideal if this were clearly stated

**Response**: We thank the reviewer for this suggestion, concomitant is replaced by same-sign

Line 521: Suggest changing to: "We find that forced perturbations to pO2 and temperature will
strongly exceed those associated with the natural system…"

**Response**: We thank the reviewer for this suggestion, implemented as suggested.

# Reviewer 2

The paper examines the effects of warming and deoxygenation on marine ecosystems by analyzing
the temperature sensitivity and oxygen requirements of metabolic rates. Utilizing CESM-LE, the
research explores the natural variability and anthropogenic impacts on the support for aerobic
metabolisms in marine ecosystems over various timescales. The study emphasizes that future climatic
changes will intensify the challenges faced by marine organisms, driving them toward their
physiological thresholds and heightening the vulnerability of marine ecosystems to extreme events.

The manuscript is well-written, and the line of thought is clear. I believe this paper is of interest to the
general audience of Biogeosciences. I only have very minor technical and clarification questions.

**Response**: We thank the reviewer for his/her well considered comments and suggestions.

L112: This equation should be labeled as Eq. 1

**Response**: This entire session is reformulated to show all equations explicitly as suggested.

L113 – 127: The definitions of $E_O$, $E_D$, and $E_S$, are not clear in this section. I suggest bringing the Eq.
in L123 earlier.

**Repones**: This entire session is reformulated to show equations explicitly as suggested.

**Line 105 – 131**: " Deutsch et al. (2015) formalized these concepts into a quantity termed the
"Metabolic Index ($\Phi$)", which is defined as the ratio of oxygen supply to an organism's resting
metabolic demand. Oxygen supply is parameterized according to a biomass-dependent scaling of $pO_2$,
capturing variation in the efficiency with which organisms acquire and utilize $O_2$. This can be
expressed as $S = \hat{\alpha}_s B^\sigma pO_2$, where $\hat{\alpha}_s$ represent gas transfer between an organism and its
environment and $B^\delta$ is the scaling of supply with biomass, $B$ (Piiper et al., 1971). Gas supply is
represented as an Arrhenius function;

$$\hat{\alpha}_s = \alpha_s exp\{\frac{-E_s}{K_B}\left[\frac{1}{T} - \frac{1}{T_{ref}}\right]\} \tag{1}$$

Resting metabolic demand is also expressed using the Arrhenius equation as

$$D = \alpha_D B^\delta exp\{\frac{-E_d}{K_B}\left[\frac{1}{T} - \frac{1}{T_{ref}}\right]\}, \tag{2}$$

where $\alpha_D$ is a species-specific basal metabolic rate, $E_d$ (eV) is the temperature dependence of oxygen
supply, T is temperature, $T_{ref}$ is the reference temperature (15°C), and $k_B$ is the Boltzmann constant
(Gillooly et al., 2001). Gas transfer is kinematically slow at low temperatures, and hence organism
viability can be limited by the energy to acquire oxygen at low temperatures, thus $E_o$ varies with
temperature. Here we account for this by adding the temperature dependence ($dE_o/dT$) to $E_o$ in
equations above ($E_o + \frac{dE_o}{dT}(T - T_{ref})$), using the mean value of $dE_o/dT = 0.022$ eV consistent with
Deutsch et al. (2020). The Metabolic Index can thus be written as the ratio of $S/D$:

$$\Phi = \frac{\alpha_s}{\alpha_D}\frac{B^\sigma}{B^\delta}\,pO_2\,exp\{\frac{-E_s}{K_B}\left[\frac{1}{T} - \frac{1}{T_{ref}}\right] + \frac{E_d}{K_B}\left[\frac{1}{T} - \frac{1}{T_{ref}}\right]\},$$

$$= A_o B^{\sigma-\delta} pO_2\,exp\{\frac{E_d - E_s}{K_B}\left[\frac{1}{T} - \frac{1}{T_{ref}}\right]\},$$

$$= A_o pO_2\,exp\{\frac{E_o}{K_B}\left[\frac{1}{T} - \frac{1}{T_{ref}}\right]\}, \qquad\qquad (3)$$

where $A_o = \alpha_S/\alpha_D$ (1/atm) is the hypoxic tolerance, $E_o = E_d - E_s$ ($E_s$ is the temperature dependence of
oxygen supply) (Deutsch et al., 2015; Penn et al., 2018). The exponent, $\varepsilon = \sigma - \delta$, is the allometric
scaling of the supply to demand ratio with biomass, is typically near zero. Therefore, in the analysis
that follows, we presume unit biomass and thus neglect potential impacts of variations in biomass."

L241: Could you comment on the negative bias in $pO_2$ in CESM-LE at 200 and 500 meters? This bias
is mainly due to limitations in biogeochemistry or physical circulation. How does this bias project to
future scenarios?

**Response:** This is a documented CESM bias. We will provide a description of the sources of the bias.

**Line 261 – 264**: "This CESM $pO_2$ bias is common among coarse-resolutions ocean models and it is
attributed to a sluggish circulation and hence weak ventilation (Long et al., 2016). These differences
ultimately matter most near the hypoxic zones and at the boundaries of habitable zones like the
Oxygen Minimum Zones (OMZs)."

L269: OMZ = Oxygen Maximum Zone? This has not been defined in the paper.

**Response**: Thanks for point this out, corrected

L278: How do you calculate the natural variability? 1σ uncertainty of the period 1920 to 1965?

**Repones**: Yes, natural variability is calculated as 1σ uncertainty of the period 1920 to 1965, now
stated explicitly.

L309: Curious if you compared temperature and $pO_2$ trend between CESM-LE and observation. I am
wondering if the CESM-LE shows reasonable trend. Any trend bias in CESM-LE here could project
bias in future scenarios.

**Response**: We did not compare CESM-LE and observations in this study.

L367: Texts on the left of the bottom row should indicate a trend (difference between 2020–2099 and
1920–1965)

**Response**. No, these plots show a $pO_2$-temperature regression at 50 m, 200 m and 500 m, the top row
is the natural climate (1920 – 1965) and bottom row, the forced trend (2020 – 2099).

CESM-LE seems to suggest deoxygenation has started only since ~2000. Observation data, however,
support an earlier onset of ocean deoxygenation. Could you comment on this?

**Response**: Thanks for pointing this out, this reflect CESM's underestimation of deoxygenation with
warming which also came in the above comment.

**Reviewer ##3**

In this manuscript, the authors use a synthesis of empirical data and ESM large ensemble to assess the
influence of both oxygen and temperature in determining habitat suitability for a series of ecotypes in
the surface and subsurface ocean, and they study how these factors, and their interaction, change
distribution of these ecotypes under climate change. The study is compelling, well thought out, and
very well written. It was truly an enjoyable read.

The only pointed criticism I have is that it is missing some context on the empirical data used.
Although the dataset is referenced in the manuscript, some added text on how it was synthesized and
broad description of types of species included, their ecological role, and how the values used were
obtained would be helpful. Rough data distribution and possible geographical biases could also be
mentioned.

Other than that, any comments I have are very minor (some cosmetic) and I would recommend this
manuscript for publication with minor revisions. Specific comments are mentioned below.

The only pointed criticism I have is that it is missing some context on the empirical data used.
Although the dataset is referenced in the manuscript, some added text on how it was synthesized and
broad description of types of species included, their ecological role, and how the values used were
obtained would be helpful. Rough data distribution and possible geographical biases could also be
mentioned.

**Response**: We thank the reviewer for his/her well considered comments and suggestions.
We added more details on the physiological datasets we used.
**Line 153 – 159**: "We make use of a dataset describing physiological parameters for a collection of 61
marine ecotypes spanning a range of ecological hypoxic tolerances ($A_c$) and temperature sensitivities
($E_o$) (Penn et al., 2018; Deutsch et al., 2020, Figure 1a). The 61 species span benthic and pelagic
habitats across four phyla in all ocean basins (Arthropoda, Chordata, Mollusca, and Cnidaria). The
dataset include 28 malacostracans, 21 fishes, three bivalves and cephalopods, two copepods, and one
each for gastropods, ascidians, scleractinian corals, and sharks with body mass spans of eight orders
of magnitude (Penn et al., 2018)."
L109: check the exponent on B

**Reponses**:  This was indeed a typo and it is corrected.

L147: add more information about the dataset used

**Response**: We added more details on the physiological datasets we used.
**Line 153 – 159**: "We make use of a dataset describing physiological parameters for a collection of 61
marine ecotypes spanning a range of ecological hypoxic tolerances ($A_c$) and temperature sensitivities
($E_o$) (Penn et al., 2018; Deutsch et al., 2020, Figure 1a). The 61 species span benthic and pelagic habitats across four phyla in all ocean basins (Arthropoda, Chordata, Mollusca, and Cnidaria). The
dataset include 28 malacostracans, 21 fishes, three bivalves and cephalopods, two copepods, and one
each for gastropods, ascidians, scleractinian corals, and sharks with body mass spans of eight orders
of magnitude (Penn et al., 2018)."

Figure 1: the blue star is really hard to see, consider using a different color

Figure 5: Perhaps add a label with the variables to the left to make interpretation easier?

**Response**: A variable description added on the left of figure 5

L376: Figure 8?

**Response**: This was as indeed a typo, corrected

Figure 8: could add title with the depth on panels a-c

**Response**. Thanks for the suggestion, title added.

L479: remove "in the surface ocean"

**Response**: Removed.

---

## Author Response (AR2)

**Rebuttal report**

Precious Mongwe[1,2], Matthew Long[3], Takamitsu Ito[4,] Curtis Deutsch[5], and

Yeray Santana-Falcón[6]

[1]Southern Ocean Carbon Climate Observatory (SOCCO), CSIR, Cape Town, South Africa

[2]National Institute for Theoretical and Computational Sciences (NITheCS), Cape Town, South Africa

[3]Oceanography Section, Climate and Global Dynamics Laboratory, National Center for Atmospheric Research, Boulder, CO, United States of America

[4]School of Earth and Atmospheric Sciences, Georgia Institute of Technology, Atlanta, Georgia United States of America

[5]Department of Geosciences, Princeton University, Princeton, NJ, United States of America

[6]CNRM, Université de Toulouse, Météo-France, CNRS, Toulouse, 31057, France

**# Editor**

**Comment**: In general the authors addressed the concerns and suggestions of the 3 reviewers. The manuscript still has a number of phrases that need correcting to improve understanding or grammar. Here are a few"

**Response**: We thank the editor for pointing out these errors and we have taken this opportunity to go through the manuscript, correcting these errors and other typographical and grammatical mistakes

**Comment**: Line 27 - what does "emerges sooner than independently from natural variability" imply? There needs to be another word inserted.
**Response**: We corrected this sentence, here is the revised version.
Line 27: "Further, the joint temperature-oxygen anthropogenic signal emerges sooner than temperature and oxygen independently from natural variability."

**Comment**: Line 74 - insert "future" after "will"
**Respond**: Excecated as suggested

**Comment**: Line 156 Change "Account" to "Accounting"
**Respond**: Excecated as suggested

Line 499- Change to "Sea-ice melting supports"

**Respond**: We could not find this sentence at the suggested line number, but we corrected a similar error in lines 396 to 397. We presume the editor may have made a reference mistake.
_**line 396 – 398**_: "Additionally, sea ice melt support an increase in $pO_2$, as gas exchange becomes more effective at restoring equilibrium oxygen concentrations."

**Comment**: Line 569 - insert "of" before metabolic resilience
**Respond**: Excecated as suggested

**Comment**: Lines 543-544 - While the warmer tropics support only 10-20 of the 61 ecotypes - it

45    should be noted that marine species diversity is well established to be: Tropics>Temperate>Polar
46    waters.""""

47

48    **Response**: Unfortunately, we could not find this sentence or a related sentence close to this line
49    reference. We found a similar sentence in lines 511 - 513. However, we could not understand how the
50    suggested correction related to the text as it would change the meaning of the sentence. Nevertheless,
51    we validate that lines 511-513 is grammatically correct, and we agree with the meaning of the
52    sentence.
53    _**Line 511-513**_: "The warmest parts of the surface ocean, the tropical oceans, can only support about
54    10-20 (~ 30%) of the 61 ecotypes while cooler regions in the extra tropics have nearly 100%
55    viability."

56

57    In addition to these changes, we also went through the manuscript and clean up other minor errors as
58    shown in the track changes document